

# Technical note: Quantification of flow field variability using intrinsic random function theory

**Ching-Min Chang[1], Chuen-Fa Ni[1], Chi-Ping Lin[2], and I-Hsian Lee[2]**

[1]Graduate Institute of Applied Geology, National Central University, Taoyuan, Taiwan

[2]Center for Environmental Studies, National Central University, Taoyuan, Taiwan

**Correspondence:** Chuen-Fa Ni (nichuenfa@geo.ncu.edu.tw)



**Abstract.** Much of the stochastic analysis of flow field variability in heterogeneous
aquifers in the literature assumes that the parameters in the associated stochastic flow
equation are weakly (second order) stationary. On this basis, the spectral
representation approach can then be used to quantify the variability of the flow fields
given known covariance functions of the input parameters. However, the condition of
second-order stationarity is rarely encountered in nature and is difficult to verify using
the limited experimental data available. The purpose (or novelty) of this work,
therefore, is to develop a new framework for modeling the variability of the flow
fields that generalizes the stochastic theory that applies to stationary second-order
random input parameters to intrinsic (nonstationary) random input parameters. In this
work, the log hydraulic conductivity and log aquifer thickness are assumed to be
intrinsic random functions for flow through heterogeneous confined aquifers of
variable thickness. On this basis, semivariograms of depth-averaged hydraulic head
and integrated specific discharge fields are developed to characterize the variability of
flow fields. The application of the proposed stochastic theory to the case where the
variability of a random input parameter can be characterized by a linear
semivariogram model is provided.
**1  Introduction**



Much of the literature on solving the stochastic groundwater flow problem
assumes that the covariance functions of the random input parameters in the
corresponding stochastic differential equation for groundwater flow can be
characterized by spatial covariance functions. Based on these known covariance
functions of parameters, the variability of flow fields in heterogeneous aquifers
can then be represented by the covariances of hydraulic head and specific
discharge using the spectral representation approach (e.g., Dagan, 1989; Gelhar,
1993; Zhang, 2002; Rubin, 2003). It is important to recognize that the approach
is built on the assumption that the random processes of the input parameters are
second order stationary, so they can be represented by a covariance function.
The question arises: can the statistics of the flow field be determined if it is not
possible to identify the covariance function of the input parameter from the
available data or if the covariance functions of the parameter do not exist?
In many practical applications, the experimental variance of a random variable
(function) sampled from a field increases with the size of the field (e.g., Desbarats and
Bachu, 1994; Molz et al., 2004; Dell'Oca et al., 2020). This means that the data have an
almost unlimited scattering capacity and cannot be properly described by ascribing a
finite a priori variance to them. This implies that the second-order stationarity



hypothesis does not appear to be suitable and that the approach assuming spatial
variation of input parameters characterized by a spatial covariance function in the
treatment of stochastic models of groundwater flow is not appropriate.

But even if there is no finite a priori variance, the spatial increments of a random

function may still have a finite variance. Note that the random function that obeys the
intrinsic hypothesis (Matheron, 1965, 1971), i.e., the assumption that the increments of
the random function are weakly stationary, is called the intrinsic (nonstationary) random
function. In this case, the variability of a nonstationary random function can be
characterized by its semivariogram. This implies that it might be possible to determine
the characteristics of the random flow fields based on the known semivariogram of
the random input parameter from the field data for the case of a nonstationary process
of the input parameter. It is clear that the intrinsic hypothesis is weaker than the
second-order stationarity hypothesis.

According to Yaglom (1987) and Christakos (1992), an intrinsic function and

its semivariogram admit a spectral representation. From these spectral
representations, the associated stochastic groundwater flow equation can be
solved in the wavenumber domain. Therefore, a spectral relationship between the
wavenumber spectra of the input parameter fluctuations and the spectra of the
output fluctuations can be obtained based on the solution of the stochastic



equation. This means that, given intrinsic semivariograms of the input parameters,
the variability of the flow fields can be characterized by the semivariograms of
the hydraulic head and the specific discharge fields using the spectral
representation approach. In other words, it is possible to establish stochastic
theories to characterize the variability of the flow fields without considering the
hypothesis of second-order stationarity for the random input parameters, which is
the goal of this study.

This work develops a general stochastic framework for quantifying the variability

of flow fields by semivariograms of depth-averaged hydraulic head and integrated
specific discharge for essentially horizontal steady groundwater flow through a
heterogeneous confined aquifer of variable thickness. It is assumed that the random
input parameters appearing in the associated stochastic differential equation, such as
the log hydraulic conductivity and the log thickness of the confined aquifer, are
intrinsic random functions and therefore nonstationarity in the depth-averaged head
and integrated discharge. This work shows how to develop a stochastic modeling
framework for quantifying the variability of the flow fields given semivariograms of
the random input parameters, which, to our knowledge, has not been presented in the
literature before. An application of the proposed stochastic theories to the case where
the variability of a random input parameter can be characterized by a linear
semivariogram model is given.

## 2    Statement of the problem


In many practical situations, a variable measured on small samples over very short
distances may exhibit very large variations over those distances. To get around this
phenomenon, a variable is often measured as an average over a given volume or area
rather than at a point. This means that in reality the field data are never collected at a
single point, but always include support with finite dimensions, so that the
semivariogram over the sample support can no longer be considered a point
semivariogram (the theoretical semivariogram). Note that the theoretical
semivariogram $\gamma(\boldsymbol{h})$ defined at point $\boldsymbol{x}$ associated with a pointwise support can be
defined as
$$\gamma(\boldsymbol{\xi}) = \frac{1}{2} Var[Z(\boldsymbol{x}+\boldsymbol{\xi}) - Z(\boldsymbol{x})],$$    (1)
In Eq. (1), $Z(\boldsymbol{x})$ is a random function.

It can be shown that the semivariogram of an intrinsic random function within a

volume $\forall$ is related to the point-theoretical semivariogram by the formula (e.g.,
Matheron, 1971; Journel and Huijbregts, 1978):
$$\gamma_{\forall}(\boldsymbol{\xi}) = \frac{1}{\forall^2} \int_{\forall} d\boldsymbol{x} \int_{\forall} \gamma(\boldsymbol{\xi}+\boldsymbol{x}-\boldsymbol{x}')d\boldsymbol{x}' - \frac{1}{\forall^2} \int_{\forall} d\boldsymbol{x} \int_{\forall} \gamma(\boldsymbol{x}-\boldsymbol{x}')d\boldsymbol{x}',$$    (2)
where $\gamma_{tr}(x)$ is the transformed semivariogram and $\gamma(x)$ is the theoretical semivariogram
defined in Eq. (1). Matheron (1971) points out that Eq. (2) holds for any intrinsic
random function, even if the covariance function does not exist.
This work presents a stochastic analysis of flow through heterogeneous confined
aquifers of variable thickness (see Appendix A). The variability of the flow results
from the variation of the random input parameters, such as the log hudraulic
conductivity and the log thickness of the confined aquifer. In this work, the log
conductivity and log aquifer thickness are considered as spatially intrinsic random
functions whose semivariogram can be represented by Eq. (2). In addition, the
variation of depth-averaged hydraulic head and integrated specific discharge can be
described by the perturbation equations (A3) and (A4), respectively. The spectral
representation approach is used to develop the semivariograms of depth-averaged
hydraulic head and vertically integrated specific discharge to quantify the variability
of the flow fields.

**3    Theoretical developments of semivariograms of flow fields**

Given the assumption that $f$ and $\beta$ in Eq. (A3) satisfy the intrinsic hypothesis, the intrinsic
random functions $f$ and $\beta$ each admit a spectral representation of the form (Yaglom, 1987;



Christakos, 1992),

$$f(x_1, x_2) = \int\limits_{-\infty}^{\infty}\int\limits_{-\infty}^{\infty} \frac{\exp[i(w_1 x_1 + w_2 x_2)] - 1}{i\sqrt{w_1^2 + w_2^2}} dZ_{Sf}(w_1, w_2),$$    (3a)

$$\beta(x_1, x_2) = \int\limits_{-\infty}^{\infty}\int\limits_{-\infty}^{\infty} \frac{\exp[i(w_1 x_1 + w_2 x_2)] - 1}{i\sqrt{w_1^2 + w_2^2}} dZ_{S\beta}(w_1, w_2),$$    (3b)

where the $w_i$ are the components of the wavenumber vector $w$ ($= (w_1, w_2)$) and $Sf(w_1, w_2)$
and $S\beta(w_1, w_2)$ are stationary spatial random processes with uncorrelated complex
Fourier increments $dZ_{Sf}(w_1, w_2)$ and $dZ_{S\beta}(w_1, w_2)$, respectively. Due to the property of the
linearity of the driving forces in Eq. (A3), the depth-averaged head perturbation can
alternatively be decomposed into two parts as
$h(x_1, x_2) = h_f(x_1, x_2) + h_\beta(x_1, x_2),$    (4a)
where $h_f$ represents the head fluctuation in response to the change in log hydraulic
conductivity, while $h_\beta$ represents the head fluctuation in response to the change in log
thickness of the aquifer. Without any restrictions, each component of the depth-averaged
head perturbation in Eq. (4a) can be expressed by Fourier-Stieltjes representations
(Priestley, 1965) as follows:
$$h_f(x_1, x_2) = \int\limits_{-\infty}^{\infty}\int\limits_{-\infty}^{\infty} \Lambda_f(x_1, x_2; w_1, w_2) dZ_{Sf}(w_1, w_2),$$    (4b)





$\quad h_\beta(x_1, x_2) = \int\limits_{-\infty}^{\infty}\int\limits_{-\infty}^{\infty} \Lambda_\beta(x_1, x_2; w_1, w_2) dZ_{S\beta}(w_1, w_2)$. $\hfill$ (4c)
In Eqs. (4b) and (4c), $\Lambda_f$ and $\Lambda_\beta$ are referred to as oscillatory functions (Priestley,

1965).

$\qquad$ Introducing Eqs. (3)-(4) into Eq. (A3), the solution of Eq. (A3) is
$\quad h_f(x_1, x_2) = J\int\limits_{-\infty}^{\infty}\int\limits_{-\infty}^{\infty} \frac{w_1}{(w_1^2 + w_2^2)^{3/2}}\left\{1 - \exp[i(w_1 x_1 + w_2 x_2)] + i(w_1 x_1 + w_2 x_2)\right\}dZ_{Sf}(w_1, w_2)$, $\hfill$ (5a)
$\quad h_\beta(x_1, x_2) = 2J\int\limits_{-\infty}^{\infty}\int\limits_{-\infty}^{\infty} \frac{w_1}{(w_1^2 + w_2^2)^{3/2}}\left\{1 - \exp[i(w_1 x_1 + w_2 x_2)] + i(w_1 x_1 + w_2 x_2)\right\}dZ_{S\beta}(w_1, w_2)$. $\hfill$ (5b)
That is,
$\quad h(x_1, x_2) = J\int\limits_{-\infty}^{\infty}\int\limits_{-\infty}^{\infty} \frac{w_1}{(w_1^2 + w_2^2)^{3/2}}\left\{1 - \exp[i(w_1 x_1 + w_2 x_2)] + i(w_1 x_1 + w_2 x_2)\right\}dZ_{Sf}(w_1, w_2)$
$\qquad +2J\int\limits_{-\infty}^{\infty}\int\limits_{-\infty}^{\infty} \frac{w_1}{(w_1^2 + w_2^2)^{3/2}}\left\{1 - \exp[i(w_1 x_1 + w_2 x_2)] + i(w_1 x_1 + w_2 x_2)\right\}dZ_{S\beta}(w_1, w_2)$. $\hfill$ (5c)
The details of the development of this solution are given in Appendix B.
$\qquad$ Furthermore, making use of the spectral representation Eq. (3) and Eq. (5) in Eq.
(A4), the perturbation for the integrated specific discharge in the direction of $x_1$ (mean
flow) is given by
$\quad q_1(x_1, x_2) = q_{f_1}(x_1, x_2) + q_{\beta_1}(x_1, x_2)$, $\hfill$ (6a)





where
$$q_{f_1}(x_1, x_2) = e^{F+B} J \int\limits_{-\infty}^{\infty} \int\limits_{-\infty}^{\infty} \frac{\exp[i(w_1 x_1 + w_2 x_2)] - 1}{i\sqrt{w_1^2 + w_2^2}} \left(1 - \frac{w_1^2}{w^2}\right) dZ_{Sf}(w_1, w_2),$$ (6b)
$$q_{\beta_1}(x_1, x_2) = e^{F+B} J \int\limits_{-\infty}^{\infty} \int\limits_{-\infty}^{\infty} \frac{\exp[i(w_1 x_1 + w_2 x_2)] - 1}{i\sqrt{w_1^2 + w_2^2}} \left(1 - 2\frac{w_1^2}{w^2}\right) dZ_{S\beta}(w_1, w_2).$$ (6c)
The semivariograms of depth-averaged head can now be calculated using Eq. (5)
in Eq. (1)
$$\gamma_h(\boldsymbol{x}, \boldsymbol{y}) = \gamma_{h_f}(\boldsymbol{x}, \boldsymbol{y}) + \gamma_{h_\beta}(\boldsymbol{x}, \boldsymbol{y}),$$ (7a)
where $\boldsymbol{x} = (x_1, x_2)$, $\boldsymbol{y} = (y_1, y_2)$, and
$$\gamma_{h_f}(\boldsymbol{x}, \boldsymbol{y}) = \Xi_1(\boldsymbol{x} - \boldsymbol{y}) + r_1 \Xi_2(\boldsymbol{x}, \boldsymbol{y}) + r_2 \Xi_3(\boldsymbol{x}, \boldsymbol{y}),$$ (7b)
$$\gamma_{h_\beta}(\boldsymbol{x}, \boldsymbol{y}) = 4\left[\Omega_1(\boldsymbol{x} - \boldsymbol{y}) + r_1 \Omega_2(\boldsymbol{x}, \boldsymbol{y}) + r_2 \Omega_3(\boldsymbol{x}, \boldsymbol{y})\right],$$ (7c)
$r_1 = x_1 - y_1$, $r_2 = x_2 - y_2$. The expressions for $\Xi_1 - \Xi_3$ and $\Omega_1 - \Omega_3$ in Eq. (7) are given in the
Appendix C. Note that the random process of the spectral representation according to
Eq. (5) and the semivariogram according to Eq. (7) is called an intrinsic random
function of order 1 (Matheron, 1973).
Similarly, the application of Eq. (6) in Eq. (1) yields the semivariogram of the
integrated specific discharge in the mean flow direction of the form
$$\gamma_q(\boldsymbol{x}, \boldsymbol{y}) = \gamma_{q_f}(\boldsymbol{x} - \boldsymbol{y}) + \gamma_{q_\beta}(\boldsymbol{x} - \boldsymbol{y}),$$ (8a)
where





$\quad \gamma_{q_f}(\boldsymbol{x}-\boldsymbol{y}) = e^{2(F+B)}J\int\limits_{-\infty}^{\infty}\int\limits_{-\infty}^{\infty}\dfrac{1-\cos(w_1 r_1)\cos(w_2 r_2)}{w_1^2+w_2^2}\Big(1-\dfrac{w_1^2}{w_1^2+w_2^2}\Big)^2 S_{Sf}(w_1,w_2)dw_1 dw_2$ ,    (8b)
$\quad \gamma_{q_\beta}(\boldsymbol{x}-\boldsymbol{y}) = e^{2(F+B)}J\int\limits_{-\infty}^{\infty}\int\limits_{-\infty}^{\infty}\dfrac{1-\cos(w_1 r_1)\cos(w_2 r_2)}{w_1^2+w_2^2}\Big(1-2\dfrac{w_1^2}{w_1^2+w_2^2}\Big)^2 S_{S\beta}(w_1,w_2)dw_1 dw_2$ .    (8c)
From Eqs. (6) and (8), it can be seen that the random process for the integrated
discharge in the mean flow direction is an intrinsic random process (or an intrinsic
random function of order 0, Matheron, 1973).

To evaluate Eqs. (7) and (8), which are used to quantify the variability of flow

fields, the spectral density functions $S_{Sf}$ and $S_{S\beta}$ must be determined. It can be shown
that when the intrinsic random function has a spectral representation as in Eq. (3), the
semivariograms of the intrinsic functions $f$ and $\beta$ are related to the covariance
functions of the stationary processes $Sf$ and $S\beta$ by
$\quad \dfrac{\partial^2}{\partial r_1^2}\gamma_f(\boldsymbol{x}-\boldsymbol{y}) + \dfrac{\partial^2}{\partial r_2^2}\gamma_f(\boldsymbol{x}-\boldsymbol{y}) = C_f(\boldsymbol{x}-\boldsymbol{y})$,    (9a)
$\quad \dfrac{\partial^2}{\partial r_1^2}\gamma_\beta(\boldsymbol{x}-\boldsymbol{y}) + \dfrac{\partial^2}{\partial r_2^2}\gamma_\beta(\boldsymbol{x}-\boldsymbol{y}) = C_\beta(\boldsymbol{x}-\boldsymbol{y})$,    (9b)
where $\gamma_f$ and $\gamma_\beta$ are semivariograms of $f$ and $\beta$ functions, respectively, and $C_f$ and $C_\beta$ are
covariance functions of $Sf$ and $S\beta$ processes, respectively. The spectral density functions of
the fluctuations of $f$ and $\beta$ are then obtained by the inverse Fourier transform of $C_f$ and
$C_\beta$, respectively, i.e.,


$\quad S_{Sf}(w_1, w_2) = \dfrac{1}{(2\pi)^2} \displaystyle\int\limits_{-\infty}^{\infty}\int\limits_{-\infty}^{\infty} \exp[w_1\xi_1 + w_2\xi_2] C_f(\xi_1, \xi_2) d\xi_1 d\xi_2,$ (10a)
$\quad S_{S\beta}(w_1, w_2) = \dfrac{1}{(2\pi)^2} \displaystyle\int\limits_{-\infty}^{\infty}\int\limits_{-\infty}^{\infty} \exp[w_1\xi_1 + w_2\xi_2] C_\beta(\xi_1, \xi_2) d\xi_1 d\xi_2.$ (10b)
Equations (7) and (8), together with Eqs. (2), (9), and (10), provide the necessary
framework for quantifying the variability of the flow fields. The results can be
obtained for specific input parameter models. This line of research will be pursued in
the next section.

**4 Application**

**4.1 The linear intrinsic semivariogram**

If a volume $\forall$ is taken as a straight segment of length $L$ and the point-theoretical
semivariogram of an input parameter in Eq. (2) is considered to be described by a
linear model (e.g., Journel and Huijbregts, 1978; Bardossy, 1997; Usowicz and Lipiec,
2021), i.e.,
$\quad \gamma(\xi) = \alpha|\xi|,$ (11)
then the transformed semivariogram in Eq. (2) can be written as


$$\gamma_L(\boldsymbol{\xi}) = \frac{\alpha}{L^2} \int_L dx \int_L |\xi + x - x'| dx' - \frac{\alpha}{L^2} \int_L dx \int_L |x - x'| dx'.$$ (12)
Note that the semivariogram of a second order stationary random function is
necessarily bounded, while the semivariogram of an intrinsic random function is not.
The integration of Eq. (12) can be performed using the Cauchy algorithm (e.g.,
Matheron, 1971)
$$\gamma_L(\xi) = \frac{\alpha}{L^2} \int_{-L}^{L} (L - |x|) |\xi + x| dx - \frac{\alpha}{L^2} \int_{-L}^{L} (L - |x|) |x| dx$$
$$= \alpha \left( |\xi| - \frac{L}{3} \right) \qquad |\xi| \geq L.$$ (13)
The details of this development are given in Appendix D. This result agrees with that
of Journel and Huijbregts (1978) obtained by a different integrating approach. Note
that $\gamma_L$ in Eq. (13) reaches $-L/3$ when $\xi$ approaches zero, and that this negative value is
called the "pseudo-negative nugget effect" (Journel and Huijbregts, 1978) due to
regularization.
In this study, it is assumed that the variograms of the input parameters depend
only on the magnitude of the distance between the two points and not on its direction.
The spatial variability of the input parameters (such as the log conductivity and log
thickness of the aquifer) can be characterized by the following semivariograms
$$\gamma_{L_f}(\xi_1, \xi_2) = \alpha_f \left( |\boldsymbol{\xi}| - \frac{L}{3} \right) \qquad |\boldsymbol{\xi}| \geq L,$$ (14a)
$$\gamma_{L_\beta}(\xi_1, \xi_2) = \alpha_\beta \left( |\boldsymbol{\xi}| - \frac{L}{3} \right) \qquad |\boldsymbol{\xi}| \geq L,$$ (14b)
which represent the extension of Eq. (13) to two dimensions. In Eq. (14), $|\boldsymbol{\xi}| =$



$(\xi_1^2 + \xi_2^2)^{1/2}$.
The covariance functions of $Sf$ and $S\beta$ processes are determined from substituting Eq.
(14) into Eq. (9), respectively,
$$C_f(\xi_1, \xi_2) = \frac{\partial^2}{\partial \xi_1^2} \gamma_{L_f}(\xi_1, \xi_2) + \frac{\partial^2}{\partial \xi_2^2} \gamma_{L_f}(\xi_1, \xi_2) = \frac{\alpha_f}{\sqrt{\xi_1^2 + \xi_2^2}},$$    (15a)
$$C_\beta(\xi_1, \xi_2) = \frac{\partial^2}{\partial \xi_1^2} \gamma_{L_\beta}(\xi_1, \xi_2) + \frac{\partial^2}{\partial \xi_2^2} \gamma_{L_\beta}(\xi_1, \xi_2) = \frac{\alpha_\beta}{\sqrt{\xi_1^2 + \xi_2^2}}.$$    (15b)
From Eqs. (10) and (15), the corresponding spectral density functions of $f$ and $\beta$ are
obtained, respectively, as follows:
$$S_{Sf}(w_1, w_2) = \frac{1}{(2\pi)^2} \int_{-\infty}^{\infty} \int_{-\infty}^{\infty} \exp[w_1 \xi_1 + w_2 \xi_2] \frac{\alpha_f}{\sqrt{\xi_1^2 + \xi_2^2}} d\xi_1 d\xi_2 = \frac{\alpha_f}{2\pi} \frac{1}{\sqrt{w_1^2 + w_2^2}},$$    (16a)
$$S_{S\beta}(w_1, w_2) = \frac{1}{(2\pi)^2} \int_{-\infty}^{\infty} \int_{-\infty}^{\infty} \exp[w_1 \xi_1 + w_2 \xi_2] \frac{\alpha_\beta}{\sqrt{\xi_1^2 + \xi_2^2}} d\xi_1 d\xi_2 = \frac{\alpha_\beta}{2\pi} \frac{1}{\sqrt{w_1^2 + w_2^2}}.$$    (16b)
The semivariogram of depth-averaged hydraulic head used to quantify the
variability of the head field can then be obtained by substituting Eq. (16) into Eq. (7)
and integrating over the wavenumber range. Note that the first term on the right-hand
side of Eq. (7b) or Eq. (7c), $\Xi_1(\boldsymbol{x}\text{-}\boldsymbol{y})$ or $4\Omega_1(\boldsymbol{x}\text{-}\boldsymbol{y})$, is called the generalized covariance
function by Matheron (1973). Figure 1 shows the numerical integration result for the
generalized covariance function of depth-averaged hydraulic head $\Xi_1$, i.e., the
component of $\gamma_{h_f}$ that reflects the effect of variation in hydraulic conductivity fields,





using Eq. (16a) in Eq. (C1). The unbounded increase in the generalized covariance
function $\Xi_1$ with separation distance suggests that there is no finite depth-averaged head
variance. This implies that the variation in depth-averaged hydraulic head does not
satisfy the second-order stationarity hypothesis. Quantifying the variability in
depth-averaged head using the assumption of second-order stationarity for the input
parameter can lead to a significant underestimation of head variability for the case of
intrinsic random log-conductivity fields. It can also be shown that similar conclusions
can be drawn from the term $4\Omega_1(x-y)$ in Eq. (7c), the component of $\gamma_{h_\beta}$ reflecting the
effect of variation in the log-aquifer thickness fields, for the case of intrinsic random
log-aquifer thickness fields.

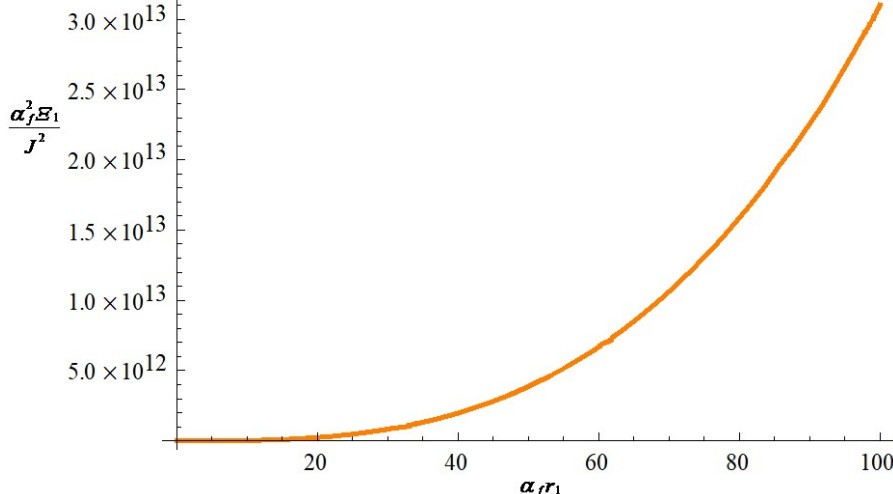


**Figure 1.** The generalized covariance function of depth-averaged hydraulic head (the
component of $\gamma_{h_f}$ that reflects the effect of variation in the log hydraulic conductivity
fields) as a function of separation distance in the mean flow direction, where $r_1 = x_1 - y_1$.





Figure 2 depicts the behavior of the generalized covariance function $\varXi_1$ as a

function of parameter $\alpha_f$ for a given separation distance $r_1$. A larger $\alpha_f$ increases the
variability of the log conductivity fields, resulting in a larger $\varXi_1$ and thus a larger
semivariogram $\gamma_{h_f}$. It can also be shown that the larger the parameter $\alpha_\beta$, the larger the
variability of the generalized covariance function $4\varOmega_1$. It can therefore be concluded
that the variability of the depth-averaged hydraulic head caused by the variation of the
log hydraulic conductivity and log aquifer thickness is larger for larger parameters $\alpha_f$
and $\alpha_\beta$.

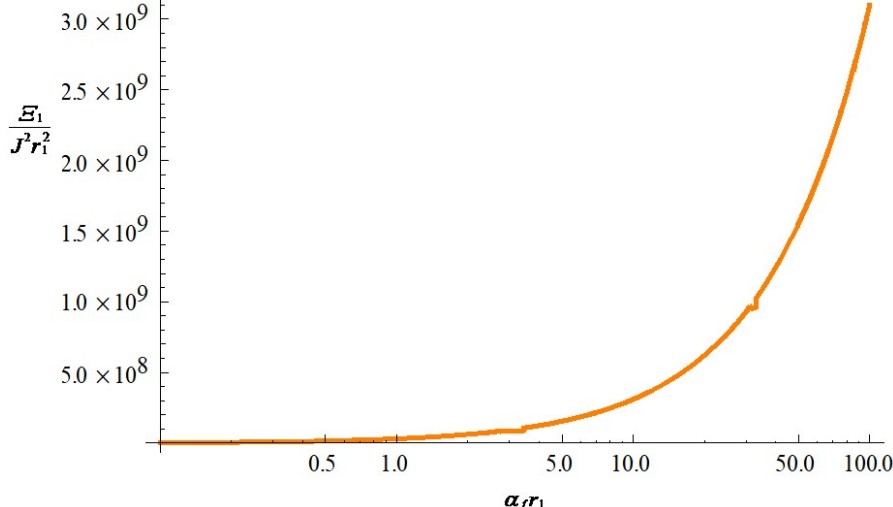


**Figure 2.** The generalized covariance function of depth-averaged hydraulic head (the
component of $\gamma_{h_f}$ that reflects the effect of variation in the log hydraulic conductivity
fields) as a function of parameter $\alpha_f$ in the mean flow direction, where $r_1 = x_1 - y_1$.

The numerical integration results for the components of the semivariogram of the
integrated specific discharge in the mean flow direction, $\gamma_{q_f}$ and $\gamma_{q_\beta}$, obtained by



substituting Eq. (16) into Eq. (8), are shown in Figs. (3a) and (3b). The unlimited
increase of the integrated discharge semivariogram with the separation distance
shown in Fig. 3 indicates that the variation of the integrated discharge process is
nonstationary. This is the result of the nonstationary process of the depth-averaged
hydraulic head caused by the intrinsic random log-conductivity and log-aquifer
thickness fields. The figure also shows that there is an increase in the semivariogram
of the integrated specific discharge in the mean flow direction with parameters $\alpha_f$ and
$\alpha_\beta$ for a given separation distance. Larger $\alpha_f$ and $\alpha_\beta$ cause greater variability in the
depth-averaged pressure fields and thus greater variability in the integrated specific
discharge fields.

**4.2 The exponential semivariogram**

It is important to note that the stationary variables always satisfy the intrinsic
hypothesis, while the opposite is not always true, since the intrinsic variable can be
nonstationary. The stochastic theory developed here to quantify the variability of the
flow fields remains valid for any second order stationary random variable. For
example, if the point theoretic semivariogram of an input parameter is chosen as
$\qquad \gamma(\xi) = \mu \left(1 - \exp\left[-\frac{|\xi|}{\lambda}\right]\right),$ $\qquad\qquad\qquad\qquad$ (17)





the transformed semivariogram over a segment of length $L$ can then be calculated
using Eq. (2) and the Cauchy algorithm (e.g., Matheron, 1971) as follows:
$$\gamma_L(\boldsymbol{\xi}) = \frac{\mu}{L^2}\int_{-L}^{L}(L-|x|)(1-\exp[-\frac{|\xi+x|}{\lambda}])dx - \frac{\mu}{L^2}\int_{-L}^{L}(L-|x|)(1-\exp[-\frac{|x|}{\lambda}])dx, \qquad (18)$$

This results in
$$\gamma_L(\xi) = \mu\frac{\lambda^2}{L^2}\{2\exp[-\frac{|\xi|}{\lambda}] - \exp[-\frac{|\xi|+L}{\lambda}] - \exp[-\frac{|\xi|-L}{\lambda}] + 2(-1+\exp[-\frac{L}{\lambda}]+\frac{L}{\lambda})\} \quad |\xi| \ge L. \quad (19)$$

For the development of Eq. (19), the reader is referred to Appendix E.

Extending Eq. (19) to two dimensions and substituting it into Eq. (9), the

covariance functions of the random input parameters ($f$ and $\beta$) can then be expressed,
respectively, as
$$C_f(\xi_1,\xi_2) = \mu_f\frac{(\exp[\frac{L}{\lambda_f}]-1)^2}{L^2}\frac{\lambda_f - \sqrt{\xi_1^2+\xi_2^2}}{\sqrt{\xi_1^2+\xi_2^2}}\exp[-\frac{\sqrt{\xi_1^2+\xi_2^2}+L}{\lambda_f}], \qquad (20a)$$

$$C_\beta(\xi_1,\xi_2) = \mu_\beta\frac{(\exp[\frac{L}{\lambda_\beta}]-1)^2}{L^2}\frac{\lambda_\beta - \sqrt{\xi_1^2+\xi_2^2}}{\sqrt{\xi_1^2+\xi_2^2}}\exp[-\frac{\sqrt{\xi_1^2+\xi_2^2}+L}{\lambda_\beta}]. \qquad (20b)$$

Using Eq. (20) in Eq. (10), it follows that the spectral density functions of the
fluctuations of $f$ and $\beta$ each have the form
$$S_{Sf}(w_1,w_2) = \frac{\mu_f}{2\pi}\frac{(\exp[\frac{L}{\lambda_f}]-1)^2}{(\frac{L}{\lambda_f})^2}\frac{\lambda_f^2(w_1^2+w_2^2)}{[1+\lambda_f^2(w_1^2+w_2^2)]^2}, \qquad (21a)$$

$$S_{S\beta}(w_1,w_2) = \frac{\mu_\beta}{2\pi}\frac{(\exp[\frac{L}{\lambda_\beta}]-1)^2}{(\frac{L}{\lambda_\beta})^2}\frac{\lambda_\beta^2(w_1^2+w_2^2)}{[1+\lambda_\beta^2(w_1^2+w_2^2)]^2}. \qquad (21b)$$


**(a)**

**(b)**

Figure 3. The components of the semivariogram of the integrated specific discharge in
the mean flow direction, (a) $\gamma_{q_f}$, reflecting the effect of variation in the log hydraulic
conductivity fields, and (b) $\gamma_{q_\beta}$, reflecting the effect of variation in the log aquifer
thickness fields, as a function of parameters $\alpha_f$ and $\alpha_\beta$ and separation distance.



Finally, substituting Eq. (21) into Eqs. (7) and (8), the semivariograms of
depth-averaged head and the semivariogram of integrated specific discharge in the
mean flow direction can now be evaluated.
The practical advantage of using the general stochastic modeling framework
developed here with the intrinsic hypothesis is a wider range of possible
semivariogram models compared to the cases with second-order stationarity. The
condition of second-order stationarity is rarely encountered in nature (e.g., Wu and Hu,
2004) and is difficult to verify using the limited experimental data available. It is
under these conditions that the presented stochastic approach has the greatest utility of
quantification of the flow field variability.

**5    Conclusions**

In this work, a general stochastic methodology is developed for quantifying the
variability of flow fields in heterogeneous confined aquifers of variable thickness. The
stochastic theories developed here, namely the semivariograms of depth-averaged
hydraulic head and integrated specific discharge used to characterize flow field
variability, can address the effects of nonstationarity due to variations in parameters
and output. The proposed stochastic theories generalize existing stochastic theory,



which applies to second order stationary random input parameters, to nonstationary
random input parameters. Stationarity in the spatial variation of soil properties is very
rarely encountered in nature. The stochastic theories developed here improve the
quantification of flow field variability in natural confined aquifers.

The results show that the introduction of intrinsic random input parameters leads

to a nonstationary process of depth-averaged hydraulic head fluctuations (an intrinsic
random function of order 1) and a nonstationary process of integrated specific
discharge fluctuations (an intrinsic random function of order 0). Application of the
stochastic theories developed here to the case where the variability of a random input
parameter can be characterized by a linear semivariogram model shows that larger
parameters $\alpha_f$ and $\alpha_\beta$ increase the variability of the depth-averaged head and thus the
variability of the integrated discharge in the mean flow direction.

## Appendix A: A steady flow through a heterogeneous confined aquifer of variable thickness

**Appendix A: A steady flow through a heterogeneous confined aquifer**
**of variable thickness**

According to Chang et al. (2021), an essentially horizontal, steady groundwater flow
through a heterogeneous confined aquifer of variable thickness can be represented as
follows:





$$\frac{\partial^2}{\partial x_i^2}\tilde{h}(x_1,x_2)+[\frac{\partial}{\partial x_i}\ln K(x_1,x_2)+2\frac{\partial}{\partial x_i}\ln b(x_1,x_2)]\frac{\partial}{\partial x_i}\tilde{h}(x_1,x_2)=0 \qquad i=1,\ 2, \qquad\qquad\text{(A1)}$$
which is the vertically integrated form of the continuity equation. In Eq. (A1), $\tilde{h}(x_1,x_2)$
is the depth-averaged hydraulic head, $K(x_1,x_2)$ is the hydraulic conductivity and $b(x_1,x_2)$
is the aquifer's thickness. From Eq. (A1), it can be seen that the variations in
hydraulic conductivity and aquifer thickness that occur affect the depth-averaged
hydraulic head. If the log conductivity and log thickness in Eq. (A1) are treated as
stochastic (random) variables, Eq. (A1) can be considered as a stochastic partial
differential equation with a stochastic output $\tilde{h}$.

Similarly, integrating the equation for specific discharge along the $x_3$-axis and

applying Leibniz's rule leads to the vertically integrated specific discharge in the $x_i$
direction as follows:
$$Q_{x_i}(x_1,x_2)=-K(x_1,x_2)b(x_1,x_2)\frac{\partial}{\partial x_i}\tilde{h}(x_1,x_2) \qquad\qquad\text{(A2)}$$

Under the influence of a uniform mean hydraulic gradient, the perturbation

equations for the depth-average hydraulic head and integrated specific discharge
associated with Eqs. (A1) and (A2) are given, respectively, by
$$\frac{\partial^2}{\partial x_i^2}h(x_1,x_2)=J[\frac{\partial}{\partial x_1}f(x_1,x_2)+2\frac{\partial}{\partial x_1}\beta(x_1,x_2)] \qquad i=1,2, \qquad\qquad\text{(A3)}$$
$$q_i(x_1,x_2)=e^{F+B}J\{[f(x_1,x_2)+\beta(x_1,x_2)]\delta_{1i}-\frac{\partial}{\partial x_i}h(x_1,x_2)\} \qquad i=1,2. \qquad\qquad\text{(A4)}$$
In Eqs. (A3) and (A4), $h$ and $q_i$ are the fluctuations of depth-average head and
integrated discharge, respectively, $J$ is the constant mean hydraulic gradient, $F$ and $B$





are the mean log conductivity and mean aquifer thickness, respectively, and $f$ and $\beta$
are the fluctuations of log conductivity and log aquifer thickness, respectively. A
detailed development of Eqs. (A3) and (A4) can be found in Chang et al. (2021).

**Appendix B: Derivation of Eq. (5)**

Since equation (A3) is linear, it can alternatively be divided into two parts as follows:
$$\frac{\partial^2}{\partial x_1^2} h_f(x_1, x_2) + \frac{\partial^2}{\partial x_2^2} h_f(x_1, x_2) = J \frac{\partial}{\partial x_1} f(x_1, x_2),$$
(B1a)

$$\frac{\partial^2}{\partial x_1^2} h_\beta(x_1, x_2) + \frac{\partial^2}{\partial x_2^2} h_\beta(x_1, x_2) = 2J \frac{\partial}{\partial x_1} \beta(x_1, x_2).$$
(B1b)

Applying Eqs. (3a) and (4b) into Eq. (B1a), it follows that
$$\frac{\partial^2}{\partial x_1^2} \Lambda_f(x_1, x_2; w_1, w_2) + \frac{\partial^2}{\partial x_2^2} \Lambda_f(x_1, x_2; w_1, w_2) = J \frac{w_1}{\sqrt{w_1^2 + w_2^2}} \exp[i(w_1 x_1 + w_2 x_2)],$$
(B2)

which is known as Poisson's equation and has a particular solution in the form
$$\Lambda_f(x_1, x_2; w_1, w_2) = J \frac{w_1}{\sqrt{w_1^2 + w_2^2}} \frac{1 - \exp[i(w_1 x_1 + w_2 x_2)] + i(w_1 x_1 + w_2 x_2)}{w_1^2 + w_2^2}.$$
(B3)

Similarly, using Eqs. (3b) and (4c), Eq. (B1b) can be written as follows:
$$\frac{\partial^2}{\partial x_1^2} \Lambda_\beta(x_1, x_2; w_1, w_2) + \frac{\partial^2}{\partial x_2^2} \Lambda_\beta(x_1, x_2; w_1, w_2) = 2J \frac{w_1}{\sqrt{w_1^2 + w_2^2}} \exp[i(w_1 x_1 + w_2 x_2)],$$
(B4)

and accordingly,





$$\Lambda_\beta(x_1,x_2;w_1,w_2) = 2J\frac{w_1}{\sqrt{w_1^2+w_2^2}}\frac{1-\exp[i(w_1x_1+w_2x_2)]+i(w_1x_1+w_2x_2)}{w_1^2+w_2^2}.$$ (B5)
Finally, substituting Eqs. (B4) and (B5) into Eq. (4), Eq. (5) is obtained.

**Appendix C: Expressions for the functions in Eq. (7)**

$$\Xi_1(\boldsymbol{x}-\boldsymbol{y}) = J^2\int_{-\infty}^{\infty}\int_{-\infty}^{\infty}\frac{w_1^2}{(w_1^2+w_2^2)^3}\left[1-\cos(w_1r_1)\cos(w_2r_2)+\frac{1}{2}(w_1^2r_1^2+w_2^2r_2^2)\right]S_{Sf}(w_1,w_2)dw_1dw_2,$$ (C1)
$$\Xi_2(\boldsymbol{x},\boldsymbol{y}) = J^2\int_{-\infty}^{\infty}\int_{-\infty}^{\infty}\frac{w_1^3}{(w_1^2+w_2^2)^3}\left[-\sin(w_1x_1)\cos(w_2x_2)+\sin(w_1y_1)\cos(w_2y_2)\right]S_{Sf}(w_1,w_2)dw_1dw_2,$$ (C2)
$$\Xi_3(\boldsymbol{x},\boldsymbol{y}) = J^2\int_{-\infty}^{\infty}\int_{-\infty}^{\infty}\frac{w_1^2w_2}{(w_1^2+w_2^2)^3}\left[-\cos(w_1x_1)\sin(w_2x_2)+\cos(w_1y_1)\sin(w_2y_2)\right]S_{Sf}(w_1,w_2)dw_1dw_2,$$ (C3)
$$\Omega_1(\boldsymbol{x}-\boldsymbol{y}) = J^2\int_{-\infty}^{\infty}\int_{-\infty}^{\infty}\frac{w_1^2}{(w_1^2+w_2^2)^3}\left[1-\cos(w_1r_1)\cos(w_2r_2)+\frac{1}{2}(w_1^2r_1^2+w_2^2r_2^2)\right]S_{S\beta}(w_1,w_2)dw_1dw_2,$$ (C4)
$$\Omega_2(\boldsymbol{x},\boldsymbol{y}) = J^2\int_{-\infty}^{\infty}\int_{-\infty}^{\infty}\frac{w_1^3}{(w_1^2+w_2^2)^3}\left[-\sin(w_1x_1)\cos(w_2x_2)+\sin(w_1y_1)\cos(w_2y_2)\right]S_{S\beta}(w_1,w_2)dw_1dw_2,$$ (C5)
$$\Omega_3(\boldsymbol{x},\boldsymbol{y}) = J^2\int_{-\infty}^{\infty}\int_{-\infty}^{\infty}\frac{w_1^2w_2}{(w_1^2+w_2^2)^3}\left[-\cos(w_1x_1)\sin(w_2x_2)+\cos(w_1y_1)\sin(w_2y_2)\right]S_{S\beta}(w_1,w_2)dw_1dw_2,$$ (C6)
$r_1 = x_1\text{-}y_1$, $r_2 = x_2\text{-}y_2$, and $S_{Sf}$ and $S_{S\beta}$ are the spectral density functions of the stationary
processes of $Sf$ and $S\beta$, respectively.




## Appendix D: Derivation of Eq. (13)


The condition for Eq. (13) that the absolute value of $\xi$ is greater than or equal to $L$ ($|\xi|$
$\geq L$) means that $\xi \geq L$ or $\xi \leq -L$. For $\xi \geq L$, the integrand of the integral Eq. (13) can be
expressed as
$$\gamma_L(\xi) = \frac{\alpha}{L^2}\int_{-L}^{0}(L+x)(|\xi|+x)dx + \frac{\alpha}{L^2}\int_{0}^{L}(L-x)(|\xi|+x)dx - \frac{\alpha}{L^2}\int_{-L}^{0}(L+|x|)(-x)dx - \frac{\alpha}{L^2}\int_{0}^{L}(L-x)xdx$$

$$= \alpha\left(|\xi| - \frac{L}{3}\right). \tag{D1}$$

For $\xi \leq -L$, the integrand of the integral Eq. (13) can be expressed as
$$\gamma_L(\xi) = \frac{\alpha}{L^2}\int_{-L}^{0}(L+x)(|\xi|-x)dx + \frac{\alpha}{L^2}\int_{0}^{L}(L-x)(|\xi|-x)dx - \frac{\alpha}{L^2}\int_{-L}^{0}(L+|x|)(-x)dx - \frac{\alpha}{L^2}\int_{0}^{L}(L-x)xdx$$

$$= \alpha\left(|\xi| - \frac{L}{3}\right). \tag{D2}$$


## Appendix E: Derivation of Eq. (19)


Analogous to Eq. (13), the integral of Eq. (18) under the condition $|\xi| \geq L$ can be
evaluated separately as the integration of Eq. (18) under the condition $\xi \geq L$ and that
under the condition $\xi \leq -L$.
For $\xi \geq L$,





$$\gamma_L(\xi) = \frac{\mu}{L^2}\int_{-L}^{0}(L+x)(1-\exp[-\frac{|\xi|+x}{\lambda}])dx + \frac{\mu}{L^2}\int_{0}^{L}(L-x)(1-\exp[-\frac{|\xi|+x}{\lambda}])dx$$

$$-\frac{\mu}{L^2}\int_{-L}^{0}(L+x)(1-\exp[\frac{x}{\lambda}])dx - \frac{\mu}{L^2}\int_{0}^{L}(L-x)(1-\exp[-\frac{x}{\lambda}])dx$$

$$= \mu\frac{\lambda^2}{L^2}\{2\exp[-\frac{|\xi|}{\lambda}] - \exp[-\frac{|\xi|+L}{\lambda}] - \exp[-\frac{|\xi|-L}{\lambda}] + 2(-1+\exp[-\frac{L}{\lambda}]+\frac{L}{\lambda})\}. \qquad (E1)$$

For $\xi \le$ -L,
$$\gamma_L(\xi) = \frac{\mu}{L^2}\int_{-L}^{0}(L+x)(1-\exp[-\frac{|\xi|-x}{\lambda}])dx + \frac{\mu}{L^2}\int_{0}^{L}(L-x)(1-\exp[-\frac{|\xi|-x}{\lambda}])dx$$

$$-\frac{\mu}{L^2}\int_{-L}^{0}(L+x)(1-\exp[\frac{x}{\lambda}])dx - \frac{\mu}{L^2}\int_{0}^{L}(L-x)(1-\exp[-\frac{x}{\lambda}])dx$$

$$= \frac{\mu^2}{L^2}\{2\exp[-\frac{|\xi|}{\lambda}] - \exp[-\frac{|\xi|+L}{\lambda}] - \exp[-\frac{|\xi|-L}{\lambda}] + 2(-1+\exp[-\frac{L}{\lambda}]+\frac{L}{\lambda})\}. \qquad (E2)$$


*Data availability*. No data was used for the research described in the article.

*Author contributions*. C-MC: Conceptualization, Methodology, Formal analysis,
Writing - original draft preparation, Writing - review & editing.
C-FN: Conceptualization, Methodology, Formal analysis, Writing - original draft
preparation, Writing - review & editing, Supervision, Funding acquisition.
C-PL: Conceptualization, Methodology, Formal analysis, Writing - original draft
preparation, Writing - review & editing.
I-HL: Conceptualization, Methodology, Formal analysis, Writing - original draft
preparation, Writing - review & editing.

*Competing interests*. The authors declare that they have no conflict of interest.




*Acknowledgements.* Research leading to this paper has been partially supported by the
grant from the Taiwan Ministry of Science and Technology under the grants MOST
110-2123-M-008-001-, MOST 110-2621-M-008-003-, and MOST
110-2811-M-008-533.

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
