# Peer review of "Technical note: Quantification of flow field variability using intrinsic random function theory"

_Hydrology and Earth System Sciences, 2023_

## Author Comment (AC1)

**Response to comments of Anonymous Referee 1**

We would like to thank the referee for the valuable comments and suggestions, which improved the quality of the paper. Below you will find our response in regard to the comments and suggestions.

**Comments to the Authors:**

This technical note provides a stochastic analysis of spatial variability in steady groundwater flow through a heterogeneous confined aquifer of variable thickness. The random log hydraulic conductivity and log thickness are assumed to be nonstationary intrinsic random functions. The article is well written, with clear objectives and development. Essentially, the authors calculate the semivariogram of the hydraulic head and the depth-hydraulic head for a linear and exponential random function model along the mean flow direction. After examining the manuscript I find that the article is small in scope given the large body of literatura in this area (even for a technical note). The authors state that there is no works on intrinsic random functions but I think this is not totally correct. For instance, Mizell et al (1982), WRR, provide a similar analysis. The authors should clearly evaluate the literatura on this topic to be able to clearly state the scientific contribution and the objective of the technical note. The work seems also limited in scope as the authors do not seem to investigate the anisotropy of the head variogram which seems to be restricted to the mean flow direction only. Also, no efforts to provide approximate solutions are given and in the end the head variogram seems to be calculated numerically. In summary, the manuscript seems correct and somehow interesting but requires more efforts in terms of literatura review and analysis of the results.

**Response**

1.

   (a) There are significant differences between the present study and Mizell et al. (1982):

      In the present study, the introduction of **the intrinsic (nonstationary) spectral representations** for the random input parameter processes (such as the **log hydraulic conductivity** and **log aquifer thickness**) leads to an intrinsic process for the perturbation of the depth-averaged head, and therefore a **nonstationary semivariogram** of the depth-averaged head is developed to quantify the variability of the depth-averaged head fields.

Mizell et al. (1982) assume that the input **log-transmissivity** processes are **stationary of the second order**, which can be described by a modified form of the Whittle covariance function. On this basis, they then evaluate the **generalized head covariance** to investigate the stationarity conditions of the head perturbation process using the theory of intrinsic random functions.

It should be noted that the head semivariogram developed here (see Eq. (7)) can be divided into two parts: one, the so-called generalized covariance, depends on the separation distance between two spatial locations, and the other depends on both the spatial location and the separation distance between two spatial locations.

The purpose of applying the theory of intrinsic random functions by Mizell et al. (1982) is to investigate under which condition the random head process can be chosen as stationary if the input log-transmissivity processes are stationary. Therefore, only the generalized head covariance is derived for the investigation, **rather than the entire head semivariogram**.

In summary, the generalized covariance of the hydraulic head developed by Mizell et al. (1982) is only valid for the random input processes of log-transmissivity with second-order stationarity. The semivariogram of the depth-averaged head proposed here applies to nonstationary and stationary random processes of the input parameters (intrinsic random processes of the logarithmic conductivity and the logarithmic thickness of the aquifer). Note that the stationary random process is the special case of the nonstationary random process.

(b) Much of the literature on stochastic subsurface hydrology dealing with the quantification of the variability of flow fields assumes that the random input parameter processes are second-order stationary and can be described by a covariance function. In the present work, the proposed stochastic theories for quantifying the variability of flow fields are developed under the assumption that the random input parameter processes are nonstationary and can be described by an intrinsic random function. The assumption of random input parameter

processes as intrinsic random processes in the derivation of the variability of flow fields has not yet been presented in the literature.

The scientific contribution of this technical note is added on page 12 (Line 181) as

"Based on the fact that the stationarity in the spatial variation of soil properties is very rare in nature, the proposed stochastic theories for quantifying the variability of flow fields generalize the existing stochastic theory, which applies to random input parameter processes with second-order stationarity, to intrinsic (nonstationary) random processes of input parameters. It is clear that the intrinsic hypothesis is weaker than the second-order stationarity hypothesis. The stochastic theories developed here improve the quantification of the variability of flow fields in natural confined aquifers of variable thickness."

2. The integral of the head semivariogram has a singularity at the origin of the wavenumber vector. Therefore, it is not possible to find a closed-form expression for the head semivariogram, which is why the integral of the head semivariogram is integrated numerically.

Many studies in the literature on stochastic hydrology are confronted with the same problem of singularity in the integral. Some of them modify the covariance function associated with the input parameter to eliminate the singularity problem (e.g., Mizell et al., 1982; Graham and McLaughlin, 1989, Li and McLaughlin, 1995). Since the use of the linear intrinsic semivariogram for the input parameter is only to provide an application for evaluating the proposed stochastic theories, we do not attempt to modify the intrinsic semivariogram of the input parameter to eliminate the singularity problem.

Mizell, S. A., Gutjahr, A. L., and Gelhar, L. W.: Stochastic analysis of spatial variability in two-dimensional steady groundwater flow assuming stationary and nonstationary heads, Water Resour. Res., 18(4), 1053-1067, 1982.

Graham, W. D. and McLaughlin, D.: Stochastic analysis of nonstationary subsurface solute transport: 2. Conditional moments, Water Resour. Res., 25(11), 2331-2355, 1989.

Li, S.-G. and McLaughlin, D.: Using the nonstationary spectral method to

analyze flow through heterogeneous trending media, Water Resour. Res., 31(3), 541-551, 1995.

3. An important assumption made in this study is the assumption of a uniform mean hydraulic head gradient in the $x_1$ direction (see Appendix A). This means that that the mean depth-averaged head is linearly related with the spatial position in the $x_1$ direction, the mean specific discharge is constant in the $x_1$ direction, but perturbations to both the hydraulic head and the discharge occur in two dimensions. The assumption of a constant gradient of mean head (or uniform mean flow) has been widely used to predict the regional groundwater flow fields in the downstream region of the aquifer in field applications (e.g., Dagan, 1989; Gelhar, 1993; Rubin, 2003). For this reason, the study focuses on quantifying the variability of the flow fields in the mean flow direction.

A note is added to page 23 (Line 364) to clarify the meaning of the assumption of a uniform mean hydraulic head gradient as

"The assumption of a uniform gradient of the mean hydraulic head means that the mean depth-averaged head is linearly related with the spatial position in the $x_1$ direction, the mean specific discharge is constant in the $x_1$ direction, but perturbations to both the hydraulic head and the discharge occur in two dimensions."

Dagan, G.: Flow and Transport in Porous Formations, Springer, New York, 1989.

Gelhar, L.W.: Stochastic Subsurface Hydrology, Prentice Hall, Englewood Cliffs, New Jersey.

Rubin, Y.: Applied Stochastic Hydrogeology, Oxford University Press, New York, 2003.

---

## Author Comment (AC2)

**Response to comments of Anonymous Referee 2**

We would like to thank the referee for the valuable comments and suggestions, which improved the quality of the paper. Below you will find our response in regard to the comments and suggestions.

**Comments to the Authors:**

This technical note provides a solution for flow field variability in the stochastic sense when the hydraulic parameters are not second order stationary, but rather can be seen as intrinsic random functions. The work develops all the equations for depth-averaged heads and flow mean and variogram. Non-stationarity is here given by variability in thickness, which is only one of the many possibilities to have such non-stationarity. As a Technical Note, everything is correct: equations, text, figures are balanced, and they provide exactly what they say.

I am not sure about the significance of this work, compared for example to another one by the same group of authors published just one year ago: Stochastic Environmental Research and Risk Assessment (2022) 36:2503-2518 https://doi.org/10.1007/s00477-021-02125-7, where the authors write specifically that they do the following: "A spectrally based perturbation approach is used to arrive at the general results for the statistics of the flow fields in the Fourier domain, such as the variance of the depth-averaged head, and the mean and variance of integrated discharge". So, the work is indeed different, but I am not sure about the significance of this additional step in the integration.

**Response**

  1. This work differs considerably from that of Chang et al. (2022):

    (a) In earlier work, published in Stochastic Environmental Research and Risk Assessment (SERRA, 2022), it was assumed that the confined aquifer with varying thickness can be approximated by an exponentially varying aquifer. This means that the variation in the thickness of the aquifer is considered **deterministic**.

    However, natural variations, such as variations in aquifer thickness caused by complex natural events, cannot be accurately predicted. Therefore, the variation in aquifer thickness is considered **random** in this work and characterized by a **nonstationary random process** with stationary

increments (i.e. an intrinsic random process). That is, the stochastic theories developed here generalize the existing stochastic theories presented in SERRA (2022) to the case of nonstationary random inputs for the logarithmic thickness of the aquifer.

To our knowledge, the consideration of the thickness of the aquifer as a random variable for the evaluation of statistics of the flow fields in a heterogeneous confined aquifers of variable thickness has not yet been presented in the literature.

(b) In earlier work (SERRA, 2022), it was assumed that the random input parameter, such as the logarithmic hydraulic conductivity field, is stationary of the second order and can be described by a covariance function. The variances of the depth-averaged head and the integrated discharge can then be obtained using the Fourier-Stieltjes representation. This means that the stochastic theories developed by Chang et al. (2022) are only valid if **the associated covariance function for the input parameter exists and is second-order stationary**.

In many practical applications, however, the covariance function of the input parameter may not exist or the second-order stationary covariance function cannot be identified from the available data. Therefore, this work develops a new approach to evaluate the flow field semivariograms using the **intrinsic (nonstationary) hypothesis for the input parameter processes** instead of the second-order stationarity hypothesis. This means the stochastic theories developed here for quantifying the flow field variability apply to nonstationary random processes of input parameters. In other words, the semivariograms of flow fields proposed here are valid even if the corresponding covariance function for the input parameter does not exist.

In summary, Chang et al. (2022) propose variances of depth-averaged head and integrated specific discharge to quantify the variability of flow fields in heterogeneous confined aquifers, where hydraulic conductivity is treated as a random variable while the variation of aquifer thickness is deterministic. In the present work, the semivariograms of depth-averaged head and integrated specific discharge are developed to quantify the variability of the flow fields, treating both hydraulic conductivity and aquifer thickness as intrinsic

(nonstationary) random processes.

2. Much of the literature on quantifying the variability of the flow fields in heterogeneous aquifers assumes that the random input parameter processes are second-order stationary and can be characterized by spatial covariance functions. However, the stationarity in the spatial variation of soil properties in heterogeneous aquifers is very rare in nature, and the covariance functions for the input parameter fields may not exist because the available data do not exhibit finite a priori variance.

The stochastic theories proposed here for quantifying the variability of flow fields generalize the existing stochastic theory to intrinsic (nonstationary) random processes of input parameters. It is clear that the intrinsic hypothesis is weaker than the second-order stationarity hypothesis. The stochastic theories developed here improve the quantification of the variability of flow fields in natural confined aquifers of variable thickness.

The significance of the present work is added on page 12 (Line 181) as follows:

> "Based on the fact that the stationarity in the spatial variation of soil properties is very rare in nature, the proposed stochastic theories for quantifying the variability of flow fields generalize the existing stochastic theory, which applies to random input parameter processes with second-order stationarity, to intrinsic (nonstationary) random processes of input parameters. It is clear that the intrinsic hypothesis is weaker than the second-order stationarity hypothesis. The stochastic theories developed here improve the quantification of the variability of flow fields in natural confined aquifers of variable thickness."